# A first survey for herbicide resistant weeds across major maize growing areas in the North Island of New Zealand

Zachary Ngow[1]*, Trevor K. James[1], Ben Harvey[2], Christopher E. Buddenhagen[1]

1 AgResearch Ltd., Ruakura Research Centre, Hamilton, New Zealand, 2 Foundation for Arable Research, Christchurch, New Zealand

* zachary.ngow@agresearch.co.nz

**Data Availability Statement:** All relevant data are within the manuscript and its Supporting Information files.

## Abstract

Weeds are increasingly documented with evolved resistance to herbicides globally. Three species have been reported as resistant in maize crops in New Zealand: *Chenopodium album* to atrazine and dicamba, *Persicaria maculosa* to atrazine and *Digitaria sanguinalis* to nicosulfuron. Despite knowledge of these cases, the distribution of these resistant biotypes is unknown. This study aimed to determine the prevalence of known resistant weeds in major maize growing areas in New Zealand, and to pro-actively screen other species for resistance. Weed seeds of broadleaf and grass species were collected from 70 randomly selected maize growing farms in the North Island in 2021–2022. Seeds were grown and treated with herbicides at recommended field rates. Atrazine-resistant *C. album* were recorded in a third of surveyed farms and nicosulfuron-resistant *D. sanguinalis* in a sixth. Half of Waikato farms and a quarter of Bay of Plenty farms (no Hawkes Bay or Wellington farms) had atrazine-resistant *C. album*. Dicamba-resistant *C. album* were not detected, nor were atrazine-resistant *P. maculosa*. Nicosulfuron resistant *D. sanguinalis* was recorded in 19% of Waikato farms, 6% of Bay of Plenty farms and 9% of Hawkes Bay farms (no Wellington farms). *Amaranthus* spp., *Fallopia convolvulus*, *Persicaria* spp., *Solanum* spp., *Echinochloa crus-galli*, *Panicum* spp. and *Setaria* spp. were not resistant to any of the herbicides tested. Twenty-nine to 52% of maize farms in the North Island are estimated to have herbicide resistant weeds. Resistance is common in maize farms in Waikato and western Bay of Plenty. Resistance is rare in southern regions, with only one instance of nicosulfuron-resistant *D. sanguinalis* and no resistant *C. album*. Most annual weeds in maize are not resistant to herbicides; although atrazine resistant *C. album* is widespread, it is currently controlled with alternative herbicides. Resistant *D. sanguinalis* appears to be an emerging problem.

## Introduction

Maize is an important crop in New Zealand, grown on an estimated 70,000 hectares as silage and grain maize to support animal agriculture [1, 2]. Growers of maize rely heavily on herbicides to control weeds. Across the world, repeated use of herbicides within a single mode of

**Funding:** This work was funded by a New Zealand Ministry for Business, Innovation and Employment grant from the Endeavour fund: C10X1806, Improved weed control and vegetation management to minimise future herbicide resistance. The funder did not play any role in the study design, data collection and analysis, decision to publish, or preparation of the manuscript.

**Competing interests:** The authors have declared that no competing interests exist.

action has led to repeated evolution of herbicide resistant weeds [3]. Nineteen species are now reported to have herbicide resistance in New Zealand [4, 5]. Almost a fifth of worldwide cases of herbicide resistance are in maize crops [6]. Here, only three species are known to be resistant in maize.

The first documented case of herbicide resistance in New Zealand was atrazine-resistant *Chenopodium album* L. from maize grown in the Waikato region in 1979 [7]. A short time later, atrazine-resistant *Persicaria maculosa* Gray was also found in Waikato maize [8]. By 1990 it was believed that most *C. album* in maize in New Zealand was atrazine-resistant [7, 9]. In the case of *P. maculosa*, an intensive eradication programme was undertaken and the resistant biotype was not believed to have spread [9]. In 2003, *C. album* populations resistant to both atrazine and dicamba were detected in eastern Waikato [10]. Many other post-emergent herbicides still controlled these populations, and growers in the area appeared to be using these alternatives [11]. Despite there being several documented cases of resistance, no systematic survey for resistant weeds in New Zealand maize crops has been conducted. In 2017, nicosulfuron-resistant *Digitaria sanguinalis* (L.) Scop populations were detected in the Bay of Plenty and Waikato [5, 12]. The current lack of knowledge regarding the prevalence of these resistant biotypes, including the recently discovered nicosulfuron-resistant *D. sanguinalis*, motivated an investigation of the prevalence of resistant weeds in maize.

Recent herbicide resistance surveys in Canterbury cereals demonstrated higher levels of resistance than had been suspected in known resistant species and detected several new species of herbicide resistant weeds [5]. We anticipated that similar detections of new resistant weed species could be observed in surveys for resistance in maize. Weed species common in maize such as *Amaranthus powelli* S.Watson, *Solanum nigrum* L. and *Echinochloa crus-galli* (L.) P. Beauv. have evolved herbicide resistance repeatedly in other parts of the world and were predicted to have a high risk of evolving resistance in New Zealand [13].

Randomized surveys can provide a reasonable estimate of the rate of resistance across farms [14]. By knowing that resistance is present, sector wide communication about the problem is possible, which can lead to farmers changing their weed management practices [15]. Repeated surveys can reveal temporal trends of resistant weeds, as demonstrated in Australian weed surveys [16, 17].

This study presents results from the first randomized survey for herbicide resistant weeds in New Zealand maize. We estimate the prevalence of herbicide resistant weeds across major maize growing regions (Bay of Plenty, Waikato, Hawke's Bay, Wairarapa).

## Materials and methods

### Sample collection

Farmer contact information was sourced from Agribase [18], and the Foundation for Arable Research (FAR) member list. Farmers were contacted in randomised order, to obtain permission for field sampling, from two databases to provide a total 70 randomly selected farms, 36 from Waikato, 16 from Bay of Plenty, 11 from Hawke's Bay and 7 from Wellington. Farmers were contacted by phone again to confirm their permission on the day we visited their farms. One field, fitting the criterion of being cropped for at least two consecutive years, was sampled for each farm.

Fields were sampled before crop harvest when most weed seed was mature. This was between late February to early March 2021 for Waikato and Bay of Plenty, and in late January 2022 for Hawke's Bay and Wellington (Wairarapa). Seed samples were collected from individual weeds (not combined across multiple plants) within surveyed fields; seed from up to ten plants per species were collected from per field, at distances of at least 20 metres apart. Our

goal to detect the presence of resistance, not to characterize the proportion of resistance in individual fields. Collecting from individual plants means that the seeds collected share a genetic history (from the 'mother') and a larger proportion of the seed sample is likely to contain resistant traits. This approach also allowed us to detect any possible cross-resistance or multiple resistance as plants share a common genetic background. Weed seeds from non-cropped areas were collected to use as susceptible control populations, and resistant control populations from prior studies of *C. album* and *D. sanguinalis* were included [5].

## Plant propagation

Seed samples were stored at 5°C for at least a month before being planted. To break dormancy, seeds of all species except *Persicaria* spp. and *Solanum* spp. were treated with a solution of 0.2% KNO₃ for 24 hours [19]. Depending on seed availability, up to 20 seeds per sample (per herbicide) were planted in rows into plastic propagation trays (Egmont RXPROPT; 34cm × 20cm × 6cm) filled with potting mix (Dalton's grass and clover mix; 33.3% coco-coir, 33.3% pumice, 33.3% bark). Trays contained four samples, with a susceptible and a resistant sample (if available) planted in the centre rows. Samples were grown in the soil medium that was kept moist (watered every 2–3 days) and kept in a temperature-regulated glasshouse at Ruakura and maintained between 18 and 25°C. Plants were grown during their normal growing seasons, between spring and autumn, with the majority planted in November.

## Herbicide treatment

Post-emergent herbicides were applied to seedlings at the 3-leaf stage for grass species and 4-leaf for the broadleaf species, according to recommendations on herbicide labels, at the highest recommended rate (Table 1). Herbicide treatments were applied with a moving belt sprayer, fitted with a single TeeJet TT11002 flat fan nozzle at 200 kPa, positioned 440 mm above the top of the trays, and calibrated to apply a water rate of 200 L/ha. The moving belt is positioned in the central third of the fan area for even coverage.

Plant survival was assessed after susceptible controls had died, usually 3–5 weeks after treatment. Seedlings with active growth that survived herbicide treatments were considered resistant. Similar to other studies [16, 17], including our earlier survey [5] we use a threshold to classify samples, those with fewer than three seedlings surviving treatment were not classified as resistant. This choice is appropriate for samples collected from individual plants. Seed samples without sufficient germination (less than 4) were classified as 'collected' but not 'tested'.

Farms with resistant weeds were mapped using the ggmap R package [20], using map tiles by Stamen Design under CC BY 4.0 and information from OpenStreetMap contributors and the OpenStreetMap Foundation under the Open Database License. Binomial 95% confidence intervals for the proportion of farms with resistant weeds were calculated with the binom R package using the 'binom.prop.test' function [21].

## Results

### Weed samples collected

The most frequently collected species were *Amaranthus* spp. (32 samples), *C. album* (112), *Fallopia convolvulus* (L.) Á.Löve (38), *Persicaria* spp. (144), *Solanum* spp. (23), *D. sanguinalis* (152), *E. crus-galli* (74), *Panicum* spp. (50) and *Setaria* spp. (65; Table 2). Together, these made up 79% of the 869 seed samples collected. The *Amaranthus* species were *A. blitum* ssp. *oleraceus* (L.) Costea, *A. deflexus* L., *A. powelli*; *Persicaria* species were *P. decipiens* (R.Br.) K.L.Wilson, *P. hydropiper* (L.) Spach, *P. lapathifolia* (L.) Delarbre, *P. maculosa*; *Solanum* species were

**Table 1. Herbicides tested.**

| Herbicide | Mode of action | Rate (g a.i ha⁻¹) | Adjuvant | Species treated |
|---|---|---|---|---|
| atrazine | Photosystem-II inhibitors [group 5] | 1500 | - | broadleaves |
| dicamba | Synthetic auxin [group 4] | 300 | - | broadleaves |
| nicosulfuron | Acetohydroxyacid synthase inhibitors [group 2] | 60 | 0.5% Bonza®️ | broadleaves and grasses |
| topramezone | 4-Hydroxyphenylpyruvate dioxygenase inhibitors [group 27] | 67.2 | - | grasses |

*S. americanum* Mill., *S. nigrum*; *Panicum* species were *P. dichotomiflorum* Michx., *P. miliaceum* L.; *Setaria* species were *S. pumila* (Poir.) Roem. & Schult., *S. verticillata* (L.) P.Beauv.

Of the 689 seed samples collected, 435 samples successfully germinated and were then treated with herbicides (Table 2). Most seed collected was from Waikato (305 samples) and Bay of Plenty (106) farms; only 24 seed samples tested were from Hawke's Bay (22) and Wellington (2), these were *Amaranthus* spp. (7 samples), *D. sanguinalis* (6), *E. crus-galli* (4), *Panicum* spp. (4) and *Setaria* spp. (3).

## Broadleaf species

Thirty-one percent of surveyed farms (95% binomial confidence interval (CI): 21–44%) had atrazine-resistant *C. album* (Table 3); this was half of surveyed Waikato farms (mean = 50%; 95% CI: 34–66%) and a quarter of Bay of Plenty farms (mean = 25%; 95% CI: 8–53%). Sixty-seven percent (CI: 58–77%) of *C. album* seed samples tested were atrazine-resistant, and 69% (CI:50–83%) of farms with *C. album* collected and tested had atrazine-resistant *C. album*. Atrazine-resistant *C. album* was found throughout the entire Waikato region and in both the western and eastern Bay of Plenty. No farms in Hawke's Bay or Wairarapa had any *C. album* present for collection. No resistance to dicamba or nicosulfuron was observed for *C. album* in Waikato or elsewhere (Table 3). No resistant populations of the broadleaf species *Amaranthus* spp. and *F. convolvulus*, *Persicaria* spp. and *Solanum* spp. were identified.

## Grass species

Overall, 13% of farms surveyed (95% CI: 6–24%) had nicosulfuron resistant *D. sanguinalis*, this included seven Waikato farms (19%; 95% CI: 9–37%), one Bay of Plenty farm (6%; 95% CI: 0–32%) and one Hawkes Bay farm (9%; 95% CI: 0–43%; Table 4). Fourteen percent (CI: 7–22%) of *D. sanguinalis* samples tested were nicosulfuron-resistant, and 26% (15–47%) of

**Table 2. Count of farms and seed samples of weed species that were tested.** This omits sites with samples that did not germinate. Seed samples were collected from individual plants within farms.

| Weed | Waikato farms | Waikato samples | Bay of Plenty farms | Bay of Plenty samples | Hawkes Bay farms | Hawkes Bay samples | Wellington farms | Wellington samples | Total farms | Total samples |
|---|---|---|---|---|---|---|---|---|---|---|
| *Amaranthus* | 9 | 17 | 4 | 4 | 2 | 7 | 0 | 0 | 15 | 28 |
| *Chenopodium* | 23 | 71 | 9 | 28 | 0 | 0 | 0 | 0 | 32 | 99 |
| *Fallopia* | 5 | 5 | 1 | 1 | 0 | 0 | 0 | 0 | 6 | 6 |
| *Persicaria* | 26 | 85 | 5 | 26 | 0 | 0 | 0 | 0 | 31 | 111 |
| *Solanum* | 4 | 9 | 1 | 4 | 0 | 0 | 0 | 0 | 5 | 13 |
| *Digitaria* | 22 | 66 | 10 | 24 | 3 | 6 | 0 | 0 | 35 | 96 |
| *Echinochloa* | 11 | 14 | 2 | 2 | 2 | 2 | 2 | 2 | 17 | 20 |
| *Panicum* | 5 | 6 | 7 | 14 | 2 | 4 | 0 | 0 | 14 | 24 |
| *Setaria* | 11 | 32 | 3 | 3 | 2 | 3 | 0 | 0 | 17 | 38 |
| **Farms Surveyed** | 36 | 305 | 16 | 106 | 11 | 22 | 7 | 2 | 70 | 435 |

**Table 3. Prevalence of herbicide resistant broadleaf weeds in 70 maize farmss in the North Island (mean percentage farms resistant out of those surveyed; 95% confidence interval percentage farms resistant out of those surveyed).** The number of farms tested are those where seed samples of that weed species were collected and tested with herbicides.

| Species | Farms Tested | atrazine | dicamba | nicosulfuron |
|---|---|---|---|---|
| *Amaranthus* spp. | 15 | 0 | 0 | 0 |
| Waikato | 9 | 0 | 0 | 0 |
| Bay of Plenty | 4 | 0 | 0 | 0 |
| Hawkes Bay | 2 | 0 | 0 | 0 |
| Wellington | 0 | - | - | - |
| *Chenopodium album* | 32 | 22 (31%) | 0 | 0 |
| Waikato | 23 | 18 (50%) | 0 | 0 |
| Bay of Plenty | 9 | 4 (25%) | 0 | 0 |
| Hawkes Bay | 0 | - | - | - |
| Wellington | 0 | - | - | - |
| *Fallopia convolvulus* | 6 | 0 | 0 | 0 |
| Waikato | 5 | 0 | 0 | 0 |
| Bay of Plenty | 1 | 0 | 0 | 0 |
| Hawkes Bay | 0 | - | - | - |
| Wellington | 0 | - | - | - |
| *Persicaria* spp. | 31 | 0 | 0 | 0 |
| Waikato | 26 | 0 | 0 | 0 |
| Bay of Plenty | 5 | 0 | 0 | 0 |
| Hawkes Bay | 0 | - | - | - |
| Wellington | 0 | - | - | - |
| *Solanum* spp. | 5 | 0 | 0 | 0 |
| Waikato | 4 | 0 | 0 | 0 |
| Bay of Plenty | 1 | 0 | 0 | 0 |
| Hawkes Bay | 0 | - | - | - |
| Wellington | 0 | - | - | - |

farms that had *D. sanguinalis* samples collected and tested had nicosulfuron-resistant *D. sanguinalis*. No resistant populations of the grass species *E. crus-galli*, *Panicum* spp. and *Setaria* spp. were identified.

## Prevalence of resistant weeds

Twenty-nine to fifty-two percent of maize farms were estimated to have herbicide resistant weeds present (mean = 40%. No resistant weeds were detected in Wellington (0%), but one farm in Hawkes Bay (9%; 95% CI: 0–42%), five in the Bay of Plenty (31%; 95% CI: 12–59%) and twenty-two in Waikato (61%; 95% CI: 44–76%; Fig 1) had resistant weeds. Six farms had both atrazine-resistant *C. album* and nicosulfuron-resistant *D. sanguinalis*, sixteen had only resistant *C. album*, six had only resistant *D. sanguinalis* and forty-two had no resistant weeds. The farms with both resistant *C. album* and *D. sanguinalis* were all located in Waikato.

## Discussion

Assuming our sample is representative, after processing nearly 700 samples from randomly selected farms, we estimated 40% (CI 29%-52%) of maize farms had resistant weeds present in the North Island. Similar rates of herbicide resistance were recorded in the South Island arable surveys, where 48% of farms had resistance (CI 37%-59%) [5]. In those surveys, multiple weed

**Table 4. Number of farms with herbicide resistant grass weeds in 70 maize farms in the North Island (percentage farms resistant out of those surveyed; 95% confidence interval percentage farms resistant out of those surveyed).** The number of farms tested are those where seed samples of that weed species were collected and tested with herbicides.

| Species | Farms Tested | nicosulfuron | topramezone |
|---|---|---|---|
| *Digitaria sanguinalis* | 35 | 10 (13%) | 0 |
| Waikato | 22 | 7 (19%) | - |
| Bay of Plenty | 10 | 1 (6%) | - |
| Hawkes Bay | 3 | 1 (9%) | 0 |
| Wellington | 0 | - | - |
| *Echinochloa crus-galli* | 17 | 0 | 0 |
| Waikato | 11 | 0 | 0 |
| Bay of Plenty | 2 | 0 | 0 |
| Hawkes Bay | 2 | 0 | 0 |
| Wellington | 2 | 0 | 0 |
| *Panicum* spp. | 14 | 0 | 0 |
| Waikato | 5 | 0 | 0 |
| Bay of Plenty | 7 | 0 | 0 |
| Hawkes Bay | 2 | 0 | 0 |
| Wellington | 0 | - | - |
| *Setaria* spp. | 17 | 0 | 0 |
| Waikato | 11 | 0 | 0 |
| Bay of Plenty | 3 | 0 | 0 |
| Hawkes Bay | 3 | 0 | 0 |
| Wellington | 0 | - | - |

species were identified as herbicide resistant for the first time in New Zealand [5]. Contrastingly, most weed species in maize are not resistant to herbicides, except for *C. album* and *D. sanguinalis*.

Atrazine resistant *C. album* is widespread (though not in the Hawke's Bay and Wellington regions) but is mostly well controlled, as several other available herbicides remain effective. Nicosulfuron resistant *D. sanguinalis* was found in Bay of Plenty, Waikato and Hawke's Bay and appears to be an emerging problem, being resistant in 13% (CI 6%-24%) of farms. It is unclear when resistance in *D. sanguinalis* first arose; the grower in the first detected case (in 2017) is known to have applied nicosulfuron for six consecutive years [22], but our data provides a useful baseline for future work looking at the rate of change in resistance in maize. Whilst there are many herbicides available for resistant *C. album* and *P. maculosa* [13], there are fewer control options for *D. sanguinalis*, though chloroacetamides (e.g. acetochlor) and HPPD-inhibitors (e.g. topramezone) are effective.

It appears that the previously reported atrazine-resistant *P. maculosa* [8] and dicamba-resistant *C. album* [10] have failed to spread, as no resistant populations were detected. The targeted control of those biotypes with other herbicides appears to have succeeded [8, 9, 11]. In the case of dicamba-resistant *C. album*, it may be that the known fitness cost associated with resistance [23] contributed to the decline of the biotype. Atrazine-resistant *C. album* was not found in Hawke's Bay or Wellington. This may indicate that resistant biotypes had never spread to those regions, or that they are well-controlled there. Those regions differ from Waikato and Bay of Plenty by their increased amount of arable cropping, as opposed to maize monocropping or dairy/maize systems. Crop rotation is believed to delay the evolution of resistance [24], which is possibly a factor in those regions' lack of herbicide resistant weeds in maize.

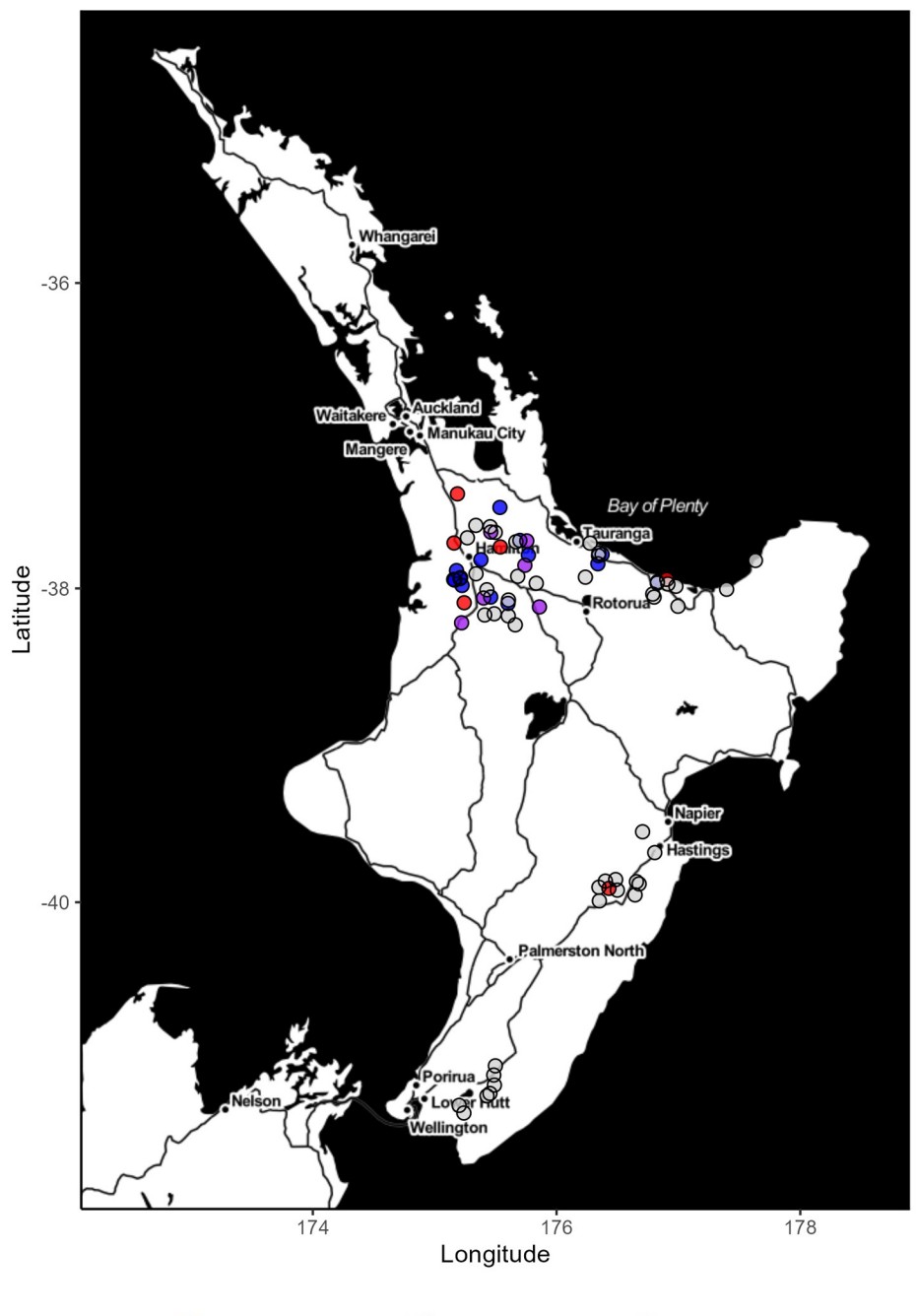

**Fig 1. Map of farms growing maize surveyed in 2021–2022 with or without resistant weeds atrazine-resistant** *Chenopodium album* **and nicosulfuron-resistant** *Digitaria sanguinalis***.** Map tiles by Stamen Design, under CC BY 4.0. Contains information from OpenStreetMap and OpenStreetMap Foundation, which is made available under the Open Database License.

With increasing awareness of the resistance problem, farmers are more willing to take precautions against resistance development [15, 25]. Herbicide resistance surveys and a free testing service briefly offered to farmers and rural professionals over a five year period [26] in Canterbury have provided direct evidence that they needed to manage their resistant weeds and encouraged a growing sense of urgency for action in farmers [5, 15]. In maize, we

identified an emerging problem of nicosulfuron-resistant *D. sanguinalis*. Here, there is a great potential for farmers to act early.

## Supporting information

**S1 Table. Herbicide resistance screening data.** This data is from weed seed samples collected from North Island, New Zealand maize crops.
(XLSX)

## Acknowledgments

We thank the farmers who allowed us to sample for weeds in their crops. Tracey Dale, Ben Wynne-Jones and Bridget Wise, Fiona Anderson and Nikita Beck assisted with collecting weed seeds and Deborah Hackell, Bridget Wise and Ben Wynne-Jones assisted with glasshouse experiments.

## Author Contributions

**Conceptualization:** Zachary Ngow, Trevor K. James, Christopher E. Buddenhagen.

**Data curation:** Zachary Ngow, Christopher E. Buddenhagen.

**Formal analysis:** Zachary Ngow.

**Funding acquisition:** Trevor K. James.

**Investigation:** Zachary Ngow, Trevor K. James, Ben Harvey.

**Methodology:** Zachary Ngow, Trevor K. James, Ben Harvey, Christopher E. Buddenhagen.

**Project administration:** Zachary Ngow, Trevor K. James.

**Resources:** Trevor K. James, Ben Harvey.

**Software:** Zachary Ngow, Christopher E. Buddenhagen.

**Supervision:** Trevor K. James, Christopher E. Buddenhagen.

**Visualization:** Zachary Ngow.

**Writing – original draft:** Zachary Ngow, Trevor K. James, Christopher E. Buddenhagen.

**Writing – review & editing:** Zachary Ngow, Ben Harvey, Christopher E. Buddenhagen.

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
