## [Decision Letter · Decision Letter 0]

7 Nov 2023

PONE-D-23-28614A first survey for herbicide resistant weeds across major maize growing areas in the North Island of New Zealand.PLOS ONE

Dear Dr. Ngow,

Thank you for submitting your manuscript to PLOS ONE. After careful consideration, we feel that it has merit but does not fully meet PLOS ONE’s publication criteria as it currently stands. Therefore, we invite you to submit a revised version of the manuscript that addresses the points raised during the review process. Very few edits are required.

We look forward to receiving your revised manuscript.

Kind regards,

Ahmet Uludag, Ph.D.

Academic Editor

PLOS ONE

Journal Requirements:

**Additional Editor Comments:**

Unfortunately, it took long time due to reviewer finding difficulty. We have three reviewers because one of them have connections with you. His review was fair but we would like to stay in safe side of ethical issues. I hope you can complete immediately and our reviewers accept reviewing.

Reviewers' comments:

Reviewer's Responses to Questions

**Comments to the Author**

1. Is the manuscript technically sound, and do the data support the conclusions?

Reviewer #1: Yes

Reviewer #2: Yes

Reviewer #3: Yes

2. Has the statistical analysis been performed appropriately and rigorously? 

Reviewer #1: Yes

Reviewer #2: Yes

Reviewer #3: Yes

3. Have the authors made all data underlying the findings in their manuscript fully available?

Reviewer #1: Yes

Reviewer #2: Yes

Reviewer #3: No

4. Is the manuscript presented in an intelligible fashion and written in standard English?

Reviewer #1: Yes

Reviewer #2: Yes

Reviewer #3: Yes

5. Review Comments to the Author

Reviewer #1: The manuscript reports the results from 2-year surveys on herbicide-resistant weeds in maize farms in New Zealand. Such surveys provide constructive and helpful information about how widespread herbicide resistance is. The manuscript is well-written and easy to read. The method deployed by the authors is valid, and the statistical analysis is robust. I only suggest a few edits to the text (please see below).

L 23 List the herbicides that you tested

L 27 Nicosulfuron-resistant

L 31 herbicide-resistant. Please check the manuscript and place a hyphen between “herbicide” and “resistant”, where both words are functioning together as an adjective.

L 35 atrazine-resistant. Please be consistent. The authors should check the entire manuscript for such inconsistencies.

L 46 Here?

L 48 Delete “from” after “Chenopodium album L.”

L 84 You should be consistent with using either American or British spelling. Earlier, you used “randomized”, but here, you used the British version.

L 85 “a total of 70”

L 101 Fertilizer?

L 102 Soil medium? Earlier, you indicated that you had grown them in the potting mix.

L 104 How about the day length? Species such as C. album require a day length of 14hday/8hnight in order to remain growing vegetatively.

L 109 Please be consistent with using “/ha” or “ha-1” (see Table 1 for example).

L 110 In Table 1, you should also include the trade name of the chemicals you used. Also, the label on atrazine recommends the addition of non-ionic surfactants. Did you use any?

L. 124 to 127 The genus of some species can be abbreviated since you did it earlier. For example, C. album.

L 136 Please be consistent with abbreviating the binomial names. If you abbreviate the genus earlier, it should be consistent for the rest of the manuscript.

L 151 “any herbicides”

L 152 Italicize “Solanum” in Table 3.

L 162 “ Echinochloa” can be abbreviated to “E.”

L 172 “22 in Waikato”

L 173 “16 had”

L 189 “Data” is a plural noun, so it should read “Our data provide”

Reviewer #2: this study provides the evidence about resistant weeds in maize fields. more dose of herbicide need to be apply to test the resistance. current data only support the weeds are not sensitive to the herbicides.

Reviewer #3: Was there the need for keeping the seed collected from the individual plants separate as in the text the results and discussion refer only to the percentage of farms with resistance

See attached file for further information

6. PLOS authors have the option to publish the peer review history of their article (what does this mean?). If published, this will include your full peer review and any attached files.

Reviewer #1: No

Reviewer #2: No

Reviewer #3: No

---

## [Author Response · Author response to Decision Letter 0]

5 Dec 2023

Authors’ response to reviewers’ comments on: A first survey for herbicide resistant weeds across major maize growing areas in the North Island of New Zealand

Overall comments

Was there the need for keeping the seed collected from the individual plants separate as in the text the results and discussion refer only to the percentage of farms with resistance.

Response: We made addition of this text to the second paragraph under the subheading ‘Sample collection’: “Our goal to detect the presence of resistance, not to characterize the proportion of resistance in individual fields. Collecting from individual plants means that the seeds collected share a genetic history (from the ‘mother’) and a larger proportion of the seed sample is likely to contain resistant traits. This approach also allowed us to detect any possible cross-resistance or multiple resistance where plants share a common genetic background.”

Introduction

Line 43 I always query the term of ‘over-use’ of herbicides in relation to the evolution of resistance. When many herbicides were first released herbicide resistance was not discussed so how can something be over-used if it is the best method of achieving something. In hindsight yes but with the available knowledge at the time no.

Change: Growers of maize rely heavily on herbicides to control weeds. Across the world, repeated use of herbicides within a single mode of action has led to the repeated evolution of herbicide resistant weeds [3].

Line 57 Were any non-random surveys done elsewhere as well? Line 60-61 basically repeats Line 57-58 but adds the non-random element.

Change: Reworded to remove repetition. To answer the question: investigations in the past of resistance did include sites around farms with resistance, but these were not done in a randomized, systematic way (see James et al. 2005, Rahman & Patterson 1987 and Rahman et al. 1983). We also tested some samples collected ‘outside’ of the survey (which were not included in the manuscript).

Line 64 This sentence doesn’t really fit anywhere. Maybe add this to end of sentence Line 68 ‘…maize in New Zealand as nor systemic random surveys have been conducted’

Change: Suggested changes made, sentence deleted.

Materials and Methods

Line 83-4 This sentence is not required, it states the obvious.

Change: Removed as suggested.

Line 91 Change ‘up-to’ to ‘up to’

Change: Suggested changes made.

Line 95-104 What time of year? Did the daylength equate to that of the weeds normal growing season?

Change: Spring-autumn. Added text: “Plants were grown during their normal growing season between spring and autumn, with the majority planted in November.”

Additional change: Subheading added for plant propagation, moved herbicide treatment to the precede the appropriate section.

Additional change - added the bolded text: Seed samples were stored at 5°C for at least a month before being planted.

Line 108 Was this a flat fan nozzle (designed for overlapping nozzles) or an even flat nozzle (banding) as a single flat fan nozzle will have different application rates across the width of the spray band without the overlap from extra nozzles.

Response: Sentence added and clarified that it is a “flat” fan nozzle. “The moving belt and trays or pots are positioned in the central third of the fan area to ensure even coverage.”

Line 110 Move Table so that it is not across two pages.

Response: The table (and other tables) now do not cross more than one page.

Line 116 Why was three chosen? Did this vary if below 20 seeds germinated. Resistance is commonly defined as greater than 20% survival, why wasn’t this definition used for consistency with other research?

Response: We clarified germination criteria for inclusion in the text. We used the same threshold in our earlier survey work, and we think this threshold is appropriate as we were not bulking samples, which is also different from the bulking approach used in Australia. 

Change: In the second paragraph under the heading “Herbicide treatment” we added: “Similar to other studies [16,17], including our earlier survey [5] we use a threshold to classify samples, those with fewer than three seedlings surviving treatment were not classified as resistant. This choice is appropriate for un-bulked samples. Seed samples without sufficient germination (less than 4) were classified as ‘collected’ but not ‘tested’.”

Results

Line 124-6 C. album, D. sanguinalis and E. crus-galli don’t need their genus spelt out as they have already had their full name written in the text 

Change: Suggested changes made 

Line 136 Same as above for D. sanguinalis and E. crus-galli

Change: Suggested changes made

Table 2 The difference between farms and samples need to be better explained. As Line 86-7 states that one paddock per farm was sampled, therefore Line 91 needs to define whether the seeds from each individual plant were kept separate (as per other NZ surveys) or bulked (as per surveys in other countries). This would explain how samples tested are greater than farms sampled. 

Response: clarification added to methods section under subtitle “Sample collection” and also to table caption.

Line 142 Delete ‘mean = 31%’ in brackets as already stated in the text

Change: Suggested changes made

Line 144 Mean of Bay of Plenty farms not provided, just says quarter while for Waikato it says both half and (50%)

Response: It was exactly half and a quarter for those but added means to clarify.

Line 144-5 What is the CI for C. album? 

Change: Added CI for proportion of resistant over tested C. album seed samples and proportion of farms with C. album resistant over tested. Also added this for D. sanguinalis in its section.

Line 150 F. convolvulus doesn’t need the genus spelt out as full name has already been written 

Change: Suggested changes made

Table 3&4 Remove second column ‘Farms surveyed’ as this just repeats for all species . Mention this at Lines 134-5 in same brackets as number of samples eg (Waikato 36 farms: 305 samples)

Change: Column removed. The number of farms sampled was specified in the materials and methods and also in Table 2 (farms surveyed). I didn’t add that as it may make that sentence too complicated. 

Line 162 E. crus-galli

Change: Suggested changes made

Discussion

Line 182 Better define this survey rather than ‘arable’ was it ‘wheat and barley farms’

Response: Mostly but not all. Arable used here. It included small numbers of other species grown for seed multiplication, e.g., clover, ryegrass (also wheat and barley is often the current crop preceded by other crop rotations.

Line 183 Change ‘weeds’ to ‘weed species’ and ‘and although’ to ‘except for C. album and D. sanguinalis.’ Start new paragraph with ‘Atrazine …’. Currently the flow of this paragraph goes from 40% resistance, not resistant, widespread resistance, this is confusing and loses the message of although resistance is common it is only in two species.

Change: Suggested changes made.

Line 194 add ‘reported previously’ and references to this sentence

Change: Suggested changes made.

Line 202-5 Move these two sentences to the first paragraph of the discussion and merge with the sentence ‘Most weed species ...’ (as suggested) as this is an important difference between these two surveys and needs to be given higher priority in the discussion 

Response: Suggested changes made.

References

1 delete ‘:a dissertation submitted in partial fulfilment of the requirements for the degree of Masters [i.e. Master] of Applied Science at Lincoln University’ and replace with ‘MAppSc Thesis’

2 remove capitals from ‘survey of maize areas and volumes’

3 Add capitals ‘Pest Management Science’

6 as per reference 2

7 as per reference 2

15 as per reference 2

19 as per reference 2, full Journal name

24 change to ‘Pesticide Resistance’

25 full Journal name

26 Is the editor’s name required in this reference?

Response: suggested changes made: replaced the text with MAppSc Thesis, fixed capitalization errors, removed the editor’s name. Additionally: italicized binomials.

A comment: In your Methods section, please provide additional information regarding the permits you obtained for the work. Please ensure you have included the full name of the authority that approved the field site access and, if no permits were required, a brief statement explaining why.

We added the text in bold: Stakeholders were contacted in randomised order, to obtain permission for field sampling, from two databases to provide a total 70 randomly selected farms, 36 from Waikato, 16 from Bay of Plenty, 11 from Hawke’s Bay and 7 from Wellington. Farmers were contacted by phone again to confirm their permission on the day we visited their farms.

A comment: We note that Figure 1 in your submission contain [map/satellite] images which may be copyrighted. All PLOS content is published under the Creative Commons Attribution License (CC BY 4.0), which means that the manuscript, images, and Supporting Information files will be freely available online, and any third party is permitted to access, download, copy, distribute, and use these materials in any way, even commercially, with proper attribution. For these reasons, we cannot publish previously copyrighted maps or satellite images created using proprietary data, such as Google software (Google Maps, Street View, and Earth). For more information, see our copyright guidelines: http://journals.plos.org/plosone/s/licenses-and-copyright.

Response: We added the attribution: “Map tiles by Stamen Design, under CC BY 4.0. Contains information from OpenStreetMap and OpenStreetMap Foundation, which is made available under the Open Database License.”

A comment: Please include captions for your Supporting Information files at the end of your manuscript, and update any in-text citations to match accordingly. Please see our Supporting Information guidelines for more information.

Response: I have included a caption as according to the guidelines: “S1 Table: Herbicide resistance screening data. This data is from weed seed samples collected from North Island, New Zealand maize crops.”

---

## [Decision Letter · Decision Letter 1]

23 Jan 2024

PONE-D-23-28614R1A first survey for herbicide resistant weeds across major maize growing areas in the North Island of New Zealand.PLOS ONE

Dear Dr. Ngow,

Thank you for submitting your manuscript to PLOS ONE. After careful consideration, we feel that it has merit but does not fully meet PLOS ONE’s publication criteria as it currently stands. Therefore, we invite you to submit a revised version of the manuscript that addresses the points raised during the review process. Your manuscript is almost done. Could you please comlete what reviewer 3 suggested.

Please submit your revised manuscript by Mar 08 2024 11:59PM. If you will need more time than this to complete your revisions, please reply to this message or contact the journal office at plosone@plos.org. Please include the following items when submitting your revised manuscript:A rebuttal letter that responds to each point raised by the academic editor and reviewer(s). You should upload this letter as a separate file labeled 'Response to Reviewers'.A marked-up copy of your manuscript that highlights changes made to the original version. You should upload this as a separate file labeled 'Revised Manuscript with Track Changes'.An unmarked version of your revised paper without tracked changes. You should upload this as a separate file labeled 'Manuscript'.If applicable, we recommend that you deposit your laboratory protocols in protocols.io to enhance the reproducibility of your results. Protocols.io assigns your protocol its own identifier (DOI) so that it can be cited independently in the future. For instructions see: https://journals.plos.org/plosone/s/submission-guidelines#loc-laboratory-protocols. Additionally, PLOS ONE offers an option for publishing peer-reviewed Lab Protocol articles, which describe protocols hosted on protocols.io. Read more information on sharing protocols at https://plos.org/protocols?utm_medium=editorial-email&utm_source=authorletters&utm_campaign=protocols.

We look forward to receiving your revised manuscript.

Kind regards,

Ahmet Uludag, Ph.D.

Academic Editor

PLOS ONE

Journal Requirements:

Additional Editor Comments.

Could you please follow points mentioned by reviewer three. Your paper is almost done.

Reviewers' comments:

Reviewer's Responses to Questions

**Comments to the Author**

1. If the authors have adequately addressed your comments raised in a previous round of review and you feel that this manuscript is now acceptable for publication, you may indicate that here to bypass the “Comments to the Author” section, enter your conflict of interest statement in the “Confidential to Editor” section, and submit your "Accept" recommendation.

Reviewer #3: All comments have been addressed

Reviewer #4: (No Response)

2. Is the manuscript technically sound, and do the data support the conclusions?

Reviewer #3: Yes

Reviewer #4: Yes

3. Has the statistical analysis been performed appropriately and rigorously? 

Reviewer #3: N/A

Reviewer #4: Yes

4. Have the authors made all data underlying the findings in their manuscript fully available?

Reviewer #3: (No Response)

Reviewer #4: Yes

5. Is the manuscript presented in an intelligible fashion and written in standard English?

Reviewer #3: Yes

Reviewer #4: Yes

6. Review Comments to the Author

Reviewer #3: Line 44 Is ‘repeated’ needed here as already used in the sentence.

Line 58-9 Sentence feels ‘clumsy’. Maybe ‘no random surveys to determine the proportion of farms with resistant weeds were conducted anywhere in New Zealand.’ Is this all crops or just maize?

Reference 3 use capitals for journal name Pest Management Science

Reference 17 remove abbreviation Crop and Pasture Science not Crop Pasture Sci

Reviewer #4: Comments attached, but just a few minor corrections suggested.

In the response to reviewers in the materials and methods section, the authors indicated that a new subheading was added for ‘ plant propagation’. This cannot be found. Everything makes sense without it, but if there is a heading or a section missing by accidental deletion, this should be included.

Line 92 The term ‘bulking’ or ‘bulked samples’ is introduced for the first time. The meaning is clear, but I wonder if this is really the most accurate or widely used term in the literature for such methods. It seems like a regional slang.

Table 2. While the description has been improved, the detail from line 93 should be included to help understand when the table stands on its own. Otherwise the description is still unintelligible.

Table 4. Entire table in a different font. The description for the heading of the current column 2 is confusing between Table 3 and 4. Both should have the same language and the description should clarify what is meant by ‘farms’, ‘samples’, ‘paddocks’ or ‘fields’, similar to including the line 93 in Table 2, so that the table can stand on its own.

7. PLOS authors have the option to publish the peer review history of their article (what does this mean?). If published, this will include your full peer review and any attached files.

Reviewer #3: No

Reviewer #4: No

---

## [Author Response · Author response to Decision Letter 1]

6 Feb 2024

Authors’ response to reviewers’ comments on: A first survey for herbicide resistant weeds across major maize growing areas in the North Island of New Zealand

Here are our changes in response to the reviewers’ comments.

A comment:

In the response to reviewers in the materials and methods section, the authors indicated that a new subheading was added for ‘ plant propagation’. This cannot be found. Everything makes sense without it, but if there is a heading or a section missing by accidental deletion, this should be included.

Change: We have put the subheading back. It may have accidentally been deleted, possibly due to merging tracked changes.

Line 58

Sentence feels ‘clumsy’. Maybe ‘no random surveys to determine the proportion of farms with resistant weeds were conducted anywhere in New Zealand.’ Is this all crops or just maize?

Change: Removed that sentence and the next. Added a sentence on line 55: “Despite there being several documented cases of resistance, no systematic survey for resistant weeds in New Zealand maize crops has been conducted.” The paragraph formerly beginning on line 57 is now merged with the previous. Removed two sentences from that paragraph reiterating the previous assumption of resistant weed distribution. Order of sentence changed to: “The current lack of knowledge regarding the prevalence of these resistant biotypes, including the recently discovered nicosulfuron-resistant D. sanguinalis, motivated an investigation of the prevalence of resistant weeds in maize.”

To answer the question: No random surveys for resistance had in fact been done before this in New Zealand. Between 2019-2023, the first surveys for resistance began in cereals, maize and vineyards. However, when a new case of resistance was detected, there were smaller local investigations of nearby farms (for instance studies of dicamba-resistant C. album included multiple farms from eastern Waikato). These studies did not describe the prevalence of those resistant weeds.

Line 72

An additional change: We changed a sentence, for clarity, to: “We anticipated that similar detections of new resistant weed species could be observed in surveys for resistance in maize.”

Line 92

The term ‘bulking’ or ‘bulked samples’ is introduced for the first time. The meaning is clear, but I wonder if this is really the most accurate or widely used term in the literature for such methods. It seems like a regional slang. 

Change: Removed the word ‘bulked’; now the sentence reads: “Seed samples were collected from individual weeds (not combined across multiple plants) within surveyed paddocks; seed from up to ten plants per species were collected from per field, at distances of at least 20 metres apart.” Additionally, changed a sentence on line 127 to: “This choice is appropriate for samples collected from individual plants.”

Table 2. 

While the description has been improved, the detail from line 93 should be included to help understand when the table stands on its own. Otherwise the description is still unintelligible. 

Change: Added to the table caption: “Count of farms and seed samples of weed species that were tested. This omits sites with samples that did not germinate. Seed samples were collected from individual plants within farms.”

Table 4. 

Entire table in a different font. The description for the heading of the current column 2 is confusing between Table 3 and 4. Both should have the same language and the description should clarify what is meant by ‘farms’, ‘samples’, ‘paddocks’ or ‘fields’, similar to including the line 93 in Table 2, so that the table can stand on its own. 

Change: Standardized font to Times New Roman. Labelled the first column in both Table 3 and 4 as ‘Farms Tested’. Added sentence to captions on Table 3 and 4: “The number of farms tested are those where seed samples of that weed species were collected and tested with herbicides.” All mentions of ‘paddocks’ have been changed to ‘fields’ for standardization. Two mentions of ‘fields’ were corrected to ‘farms’ in the descriptions of Table 3 and 4. 

Reference 3

use capitals for journal name Pest Management Science

Change: Corrected to capitals.

Reference 17

remove abbreviation Crop and Pasture Science not Crop Pasture Sci

Change: Corrected to Crop and Pasture Science.

An additional change:

We have changed the order of the authors to better reflect their contributions: Zachary Ngow, Trevor K. James, Ben Harvey and Christopher E. Buddenhagen

---

## [Decision Letter · Decision Letter 2]

13 Feb 2024

A first survey for herbicide resistant weeds across major maize growing areas in the North Island of New Zealand.

PONE-D-23-28614R2

Dear Dr. Ngow,

We’re pleased to inform you that your manuscript has been judged scientifically suitable for publication and will be formally accepted for publication once it meets all outstanding technical requirements.

Kind regards,

Ahmet Uludag, Ph.D.

Academic Editor

PLOS ONE

Additional Editor Comments (optional):

congratulations.

Reviewers' comments:

Reviewer's Responses to Questions

**Comments to the Author**

1. If the authors have adequately addressed your comments raised in a previous round of review and you feel that this manuscript is now acceptable for publication, you may indicate that here to bypass the “Comments to the Author” section, enter your conflict of interest statement in the “Confidential to Editor” section, and submit your "Accept" recommendation.

Reviewer #3: All comments have been addressed

Reviewer #4: All comments have been addressed

2. Is the manuscript technically sound, and do the data support the conclusions?

Reviewer #3: Yes

Reviewer #4: Yes

3. Has the statistical analysis been performed appropriately and rigorously? 

Reviewer #3: N/A

Reviewer #4: Yes

4. Have the authors made all data underlying the findings in their manuscript fully available?

Reviewer #3: Yes

Reviewer #4: Yes

5. Is the manuscript presented in an intelligible fashion and written in standard English?

Reviewer #3: Yes

Reviewer #4: Yes

6. Review Comments to the Author

Reviewer #3: Only comment - Line 125 needs a space between 'for' and 'sample'. Other than that all comments addressed.

Reviewer #4: (No Response)

7. PLOS authors have the option to publish the peer review history of their article (what does this mean?). If published, this will include your full peer review and any attached files.

Reviewer #3: No

Reviewer #4: No

---

## [Editor Report · Acceptance letter]

26 Feb 2024

PONE-D-23-28614R2 

PLOS ONE

Dear Dr. Ngow, 

I'm pleased to inform you that your manuscript has been deemed suitable for publication in PLOS ONE. Congratulations! Your manuscript is now being handed over to our production team.

Kind regards, 

on behalf of

Dr. Ahmet Uludag 

Academic Editor

PLOS ONE